# Molecular and Functional Characterization of Caveolae in Mixed Cultures of Human NT-2 Neurons and Astrocytes

**Jagdeep K. Sandhu** [1,2,*], **Maria Ribecco-Lutkiewicz** [1] **and Abedelnasser Abulrob** [1,3,*]

1   Human Health Therapeutics Research Centre, Building M54, National Research Council Canada, 1200 Montreal Road, Ottawa, ON K1A 0R6, Canada; mribecco@gmail.com
2   Department of Biochemistry, Microbiology and Immunology, Faculty of Medicine, University of Ottawa, 451 Smyth Road, Ottawa, ON K1H 8M5, Canada
3   Department of Cellular and Molecular Medicine, Faculty of Medicine, University of Ottawa, 451 Smyth Road, Ottawa, ON K1H 8M5, Canada
*   Correspondence: Jagdeep.sandhu@nrc-cnrc.gc.ca (J.K.S.); abedelnasser.abulrob@nrc-cnrc.gc.ca (A.A.); Tel.: +1-613-993-5304 (J.K.S.)

**Abstract:** Caveolae are plasma membrane invaginations that are enriched in cholesterol-binding proteins called caveolins. The presence of caveolae and caveolins in mixed cultures of human neurons and glia has not been investigated. Here, we sought to determine the presence of caveolae and caveolins in human NTera-2 (NT2/D1) cells, differentiated with retinoic acid into neuron-like (NT2/N) and astrocyte-like (NT2/A) cells. We found that while caveolin-3 mRNA levels remained relatively constant, caveolin-1 and -2 levels were upregulated in NT2/A and downregulated in NT2/N. No caveolin-1 immunoreactivity was detected in NT2/N. Electron microscopy revealed numerous flask-shaped invaginations (~86–102 nm in diameter) in the plasma membrane of NT2/A and NT2/N cells, while only few were detected in NT2/D1 cells. Immunoelectron microscopy localized caveolin-1 gold particles in the flask-shaped structures on plasmalemma and cytoplasmic vesicles of NT2/A cells. Furthermore, NT2/A endocytosed Alexa 488 conjugated-cholera toxin B subunit (CTX-B) through a caveolae- and clathrin-dependent pathway, whereas NT2/N endocytosed CTX-B through a caveolae-independent pathway. We have established that while NT2/A expressed functional caveolae, the molecular identity of the plasma membrane invaginations in NT2/N is unknown. The expression of caveolin proteins was differentially regulated in these cells. Taken together, our findings support the usefulness of the human NT2 model system to study the role of caveolins in neuron–glia communication, and their involvement in brain health and disease.

**Keywords:** human NT2/D1 cells; differentiation; retinoic acid; neuroglia; neuron-glia communication; endocytosis; caveolins; caveolin-2 variant; signal transduction; immunocytochemistry; immunoelectron microscopy

## 1. Introduction

Caveolae (little caves) are non-clathrin coated, 50–100 nm in diameter, omega- or flask-shaped invaginations of the plasma membrane which are highly enriched in glycosphingolipids, cholesterol and sphingomyelin. These plasma membrane pits are decorated by a family of 18–24 kDa cholesterol binding and scaffolding proteins known as caveolins. To date, three caveolin genes have been cloned and characterized in mammals, namely, caveolin-1, -2 and -3 [1]. Two isoforms of caveolin-1 (caveolin-1$\alpha$- and caveolin-1$\beta$) are known to be generated from a single transcript with alternate translation start sites. The caveolin proteins are expressed at varying levels in different cell types and tissues and perform highly specialized functions. For instance, caveolin-1 is present in most cell types, but is abundant in terminally differentiated cells such as adipocytes, fibroblasts, epithelial and type I pneumocytes. Caveolin-2 has not been extensively studied, but has a similar distribution to caveolin-1 [2], whereas caveolin-3 is highly expressed in muscle

cells (skeletal, cardiac, and smooth). In general, caveolins and the adapter proteins, cavins, form the structural backbone of caveolae and have been implicated in the stabilization and maintenance of caveolar structures [3]. Consistent with this, recombinant overexpression of caveolin-1 in caveolin-1-deficient cells, induces de novo formation of caveolae at the plasma membrane [4,5]. On the other hand, the lowering of caveolin-1 expression is associated with a reduction in the number of caveolae [6].

Caveolae have emerged as key molecular signaling platforms that have an important role in vesicular trafficking, potocytosis, lipid metabolism, cell growth, apoptosis and signal transduction [7,8]. In the past few years, there has been a tremendous increase in the list of signal transduction molecules that have been localized to caveolae and shown to directly interact with caveolins via the caveolin-scaffolding domain. These molecules include the heterotrimeric G-proteins and adenylyl cyclase, G-protein-coupled membrane receptors, Src protein tyrosine kinases, small GTPases (Ras, RhoA, Cdc42 and Rac), glycosylphosphatidylinositol (GPI)-anchored proteins, inositol 1,4,5-triphosphate receptors, protein kinase C (PKC), epidermal growth factor receptors, ATP-dependent $Ca^{2+}$-pump proteins, multiple components of the tyrosine kinase-mitogen-activated protein kinase pathway, and endothelial nitric oxide synthase (eNOS) [9]. Therefore, caveolae and caveolin proteins function as preassembled signaling platforms for integrating and transducing signals from the cell surface to the intracellular compartment.

Based on previous studies carried out on whole brain homogenates, cultured neurons and neuroblastoma cells, it was believed for a long time that caveolin proteins were absent in the brain tissue [10,11]. With the advent of new molecular genetic and imaging tools, an increasing number of studies have described the presence of all three caveolin isoforms in brain cells, including brain endothelial cells, astrocytes, microglia and neurons [12–15]. Niesman et al., have demonstrated that while caveolin-1 expression was increased in activated microglia, it was decreased and redistributed from the plasma membrane to cytoplasmic vesicles in resting microglia [15]. In an elegant study, Chow et al., have recently shown that arteriolar brain endothelial cells have abundant caveolae which play a major role in neurovascular coupling [16]. Using cellular models of neurons, caveolins were detected in differentiated PC-12 cells, dorsal root ganglion neurons [17] and hippocampal neurons [18]. Evidence has been building that supports the role of caveolin proteins in modulating neuronal function in the brain [12,17,19–21]. Of the caveolin proteins, caveolin-1 is the predominant scaffolding protein of caveolae that is known to organize a diverse array of signaling complexes to regulate many cellular processes, including proliferation and senescence [22]. Although the role of caveolin-1 has not been thoroughly investigated in the brain, it is emerging as a "gatekeeper molecule" and a major determinant of neuronal health. Recently, caveolin-1 has been shown to regulate neurogenesis, axonal growth, neurotrophic factor signaling and synaptic remodeling [23]. Caveolin-1 also participates in normal physiological aging and pathological processes in the brain. During normal aging, caveolin-1 expression was increased in all the examined brain regions of old rats and an aged human cerebral cortex, hippocampus, cerebellum and midbrain. In these studies, caveolin-1 expression was selectively localized to senescent neurons and was shown to alter amyloid precursor protein processing by promoting β-secretase activity [24]. Caveolin-1 deficient mice exhibited neurological abnormalities accompanied with loss of synapses, enhanced astrogliosis and changes in brain vasculature [25]. Taken together, it is evident that caveolin-1 can exert both beneficial and detrimental effects in the brain.

Although a large number of studies have investigated the role of caveolin proteins in human stem cell differentiation [22,26], only a few studies have studied their role in neuronal differentiation [27–29]. In the present study, we have used the well-established human NT2/D1 pluripotent stem cells that can be differentiated with retinoic acid into post-mitotic neuron-like cells (NT2/N) [30] and functional astrocyte-like cells (NT2/A) [31,32]. Similar to primary cultures, differentiated NT2/N and NT2/A cells can be maintained as pure or mixed cultures [32,33], recapitulating the in vivo neuronal environment. The main aim of this study was to identify caveolae and their structural caveolin proteins in

the mixed neuron-astrocyte cultures derived from NT2 cells at the mRNA and protein level using RT-PCR, western blot, immunofluorescence and immunoelectron microscopy. We have for the first time shown that NT2-derived NT2/A express caveolae and NT2/N express plasma membrane invaginations, caveolin proteins and a functional endocytosis machinery, evaluated by the internalization of cholera toxin B-subunit. Together, our results support the use of the human NT2 model system as an inexpensive tool to study the role of caveolins under normal physiological and pathological conditions of the brain, with the hope of improving therapeutic interventions.

## 2. Materials and Methods

### 2.1. Cell Culture

NTera-2 (NT2/D1) neuronal precursor cells (Stratagene, La Jolla, CA, USA) were grown in high glucose Dulbecco's modified Eagle's medium (Gibco BRL, Burlington, Ontario) supplemented with 10% FBS and 20 μg/mL gentamicin (complete DMEM) and differentiated as described previously [30]. Briefly, $2.0 \times 10^6$ per 75-cm$^2$ flask (Falcon, Lincoln Park, NJ, USA) were plated and treated three times a week for 4 weeks with 10 μM all-trans-retinoic acid (RA; Sigma Chemical Co., Oakville, ON, Canada). Cells were then harvested by trypsinization and transferred into 175-cm$^2$ flasks for 24 h in complete DMEM, in the absence of RA. Pure cultures of NT2/A, which consisted of ~99% astrocytes [31] and mixed cultures of NT2-N/A and pure cultures of NT2/N, were prepared as previously described [33]; loosely attached cells were gently knocked off and re-plated either in 75-cm$^2$ flasks ($8 \times 10^6$ cells for RNA and protein extraction), or in 12-well dishes ($5 \times 10^5$ cells for immunofluorescence microscopy). Cultures were fed twice weekly for 2 weeks with complete DMEM supplemented with DNA synthesis inhibitors (1-β-D-arabinofuranosylcytosine (1 μM), 5-fluoro-2′-deoxyuridine (10 μM) and uridine (10 μM), Sigma) to inhibit astrocytic proliferation, and then for one more week with complete DMEM.

### 2.2. RNA Isolation and the Reverse Transcriptase-Polymerase Chain Reaction

Total RNA was isolated using TRIREAGENT isolation kit (Molecular Research Center, Inc., Cincinnati, OH, USA), according to the manufacturer's protocol. To synthesize first strand of cDNA, 0.5 μg total RNA was reverse transcribed using 200 U of Moloney murine leukemia virus-reverse transcriptase, containing 10 pmols of oligo dT, 10 mM DTT and 1 mM dNTPs in a total volume of 20 μL. Incubation was carried out at 42 °C for 60 min. PCR amplifications (25 μL reactions) were carried out in the PTC-200 Peltier Thermal cycler (M.J Research, Watertown, MA, USA), using Promega PCR kit and 0.2 μM oligonucleotide primers for caveolin genes. Details of the PCR primers designed using Oligo™ 4.0 primer design package for each of the human caveolin genes are found in Table 1. The PCR amplification of caveolin-1, -2 and -3 genes was carried out as follows: initial denaturation at 95 °C for 2 min, followed by denaturation at 94 °C for 45 s, annealing at 55 °C for 45 s, and polymerization at 72 °C for 45 s for a total of 25–30 cycles. The housekeeping gene, β-actin, was used as a reference gene and was amplified from each sample using specific primers (Table 1). RT-PCR analysis for the alternate splice variant of caveolin-2 was carried out using 10 ng of cDNA from various cell types with the following conditions: initial denaturation at 94 °C for 5 min, followed by denaturation at 94 °C for 50 s, annealing at 60 °C for 50 s, and polymerization at 72 °C for 50 s for a total of 30 cycles, with the last cycle at 72 °C for 5 min. The PCR products were separated by electrophoresis on a 2% agarose gel, visualized on a UV transilluminator and photographed.

**Table 1.** Gene-specific primers used for RT-PCR.

| Gene | Primer Sequence | Product Size (bp) | GenBank Accession Number |
|---|---|---|---|
| Caveolin-1 | F: 5′ TCA ACC GCG ACC CTA AAC ACC 3′<br>R: 5′ TGA AAT AGC TCA GAA GAG ACA T 3′ | 561 | Z18951 |
| Caveolin-2 | F: 5′ AAGCTTTTCATGGACGACGACTCCTAC 3′<br>R: 5′ ACAATCCTGGCTCAGTTGCA 3′ | 451 | U32114 |
| Caveolin-3 | F: 5′ GAA GGA GGT CTA AAG CCA GG 3′<br>R: 5′ ATC TTA CAG GCA CGA ACA AA 3′ | 458 | NM_033337 |
| β-actin | F: 5′ GGA GCA ATG ATC TTG ATC TT 3′<br>R: 5′ CCT TCC TGG GCA TGG AGT CCT 3′ | 250 | NM_001613 |

### 2.3. Subcellular Fractionation and Isolation of Low-Density Caveolae Membranes

NT2/A cells were gown to confluence in 175-cm$^2$ flasks. Caveolin-rich membrane domains were prepared using a detergent-free method as described previously [34]. All steps were carried out at 4 °C and all buffers were supplemented with protease inhibitor cocktail (Sigma). Briefly, eight 175-cm$^2$ flasks of confluent cultures were washed with 10 mL of buffer A (0.25 M sucrose, 1 mM EDTA, and 20 mM Tricine, pH 7.8), collected by scraping in 5 mL of buffer A, pelleted by centrifugation at 1400× $g$ for 5 min (Beckman J-68). The pellets were resuspended in 1 mL of buffer A and homogenized in a Teflon glass homogenizer by 20 up/down strokes. Homogenized cells were centrifuged twice at 1000× $g$ for 10 min (Eppendorf Centrifuge 5415C) and the two postnuclear supernatant fractions were collected, pooled and overlaid on top of 23 mL of 30% Percoll solution in buffer A. The tubes were subjected to ultracentrifugation at 83,000× $g$ for 30 min in a Beckman 60Ti rotor. The pellet, representing plasma membrane fraction, was collected and then sonicated 6 times at 50 J/W per second (Fisher Sonic Dismembrator 300). The sonicated plasma membrane fraction was mixed with 50% Optiprep in buffer B (0.25 M sucrose, 6 mM EDTA, and 120 mM Tricine, pH 7.8) (final Optiprep concentration was 23%). The entire solution was placed at the bottom of the tube, overlaid with a linear 20–10% Optiprep gradient and centrifuged at 52,000× $g$ for 90 min using SW41Ti (Beckman Instruments). The top 5 mL of the gradient was collected and mixed with 50% Optiprep in buffer B, placed on the bottom of a SW41Ti tube and overlaid with 2 mL of 5% Optiprep in buffer A, and centrifuged at 52,000× $g$ for 90 min. An opaque band located just above the 5% interface was designated as the "caveolae fraction".

### 2.4. SDS-Polyacrylamide Gel Electrophoresis (SDS-PAGE) and Western Blot Analysis

Total proteins were extracted from NT2 cultures in RIPA buffer (50 mM Tris-HCL, pH 7.4, 150 mM NaCl, 2 mM EDTA, 1% Triton X-100, 0.1% SDS, 1% sodium deoxycholate) containing 1 mM PhenylMethaneSulfonyl Fluoride and 1× protease inhibitor cocktail. Alternatively, various cellular fractions obtained by subcellular fractionation were also used for Western blot. An equal amount of protein from each lysate (15 μg) or subcellular fraction (5 μg) was separated by 15% SDS-PAGE and transferred to PVDF membranes (Immobilon-P, Millipore) in 25 mM Tris-HCl, 192 mM glycine, and 25% methanol. Equal protein loading was verified by Coomassie blue staining. The membranes were blocked by incubation with 5% skim milk in TBST (10 mM Tris-HCl, pH 8.5, 150 mM NaCl, 0.1% Tween-20) and probed with anti-caveolin-1 (rabbit polyclonal, 1:500, Santa Cruz Biotechnology Inc., Santa Cruz, CA, USA) or anti-caveolin-3 antibody (mouse monoclonal, 1:500, Santa Cruz Biotechnology Inc., Santa Cruz, CA, USA) for 1 h at room temperature. Detection of the primary antibodies was achieved by using horseradish peroxidase-conjugated anti-rabbit IgG (1:5000, Sigma) or anti-mouse IgG (1:5000, Sigma). Immunoreactive bands were detected with the ECL plus the Western blotting detection system (Amersham Pharmacia Biotech, QC, Canada).

### 2.5. Sequence Analysis

The 268 bp band was cut from the gel, cleaned using the GENECLEAN SPIN kit (Q-BIOgene) and subcloned into P-Drive vector (Qiagen) as per manufacturer's instructions. Sequencing was performed using the ABI Prism 377, DNA sequencer. The database sequence search was performed using BLAST (http://www.ncbi.nlm.nih.gov).

### 2.6. Immunofluorescence Microscopy

NT2 cells ($5 \times 10^5$ in 12-well dishes) were grown on glass coverslips, washed with PBS and fixed in PEM buffer (80 mM PIPES, 5 mM EGTA and 1 mM MgCl$_2$, pH 6.8), containing 0.25% glutaraldehyde, 3.7% paraformaldehyde and 0.25% Triton X-100 or Genofix (DNA Genotek Inc.) for 10 min. Cells were rinsed 2× with PBS and non-specific binding was blocked for 20 min with universal blocking solution (DAKO diagnostics Canada Inc., Mississauga, BC, Canada). Double immunolabeling of mixed NT2-N/A cultures was carried out by incubation with anti-microtubule associated protein 2 (MAP2, 1:200, rabbit polyclonal, Chemicon International, Temecula) followed by detection with Alexa 488-conjugated goat anti-rabbit IgG (1:500, Molecular Probes Inc., Waltham, MA, USA). Cells were blocked again and then incubated with anti-glial fibrilliary acidic protein (1:100, mouse monoclonal clone GA-5, Neomarkers, Fremont, CA, USA) followed by detection with cy3-conjugated goat anti-mouse IgG (1:200, Jackson ImmunoResearch, West Grove, PA, USA). Cycling cells were detected using anti-Ki67/MIB-1, 1:50 (mouse monoclonal, Immunotech, Beckman coulter Inc., Brea, CA, USA) followed by incubation with 1:200 dilution of cy3-conjugated goat anti-mouse IgG. For detection of caveolin-1, cultures were incubated for 1 h at room temperature with anti-caveolin-1 antibody (pAb N-20, Santa Cruz Biotechnology Inc., Santa Cruz, CA, USA) at 1:100, followed by detection with cy3-conjugated goat anti-mouse IgG (1:200, Jackson ImmunoResearch, West Grove, PA, USA). Incubation with secondary antibodies was carried out for 45 min at room temperature. The specificity of the caveolin-1 antibody was demonstrated by preincubation of diluted caveolin-1 antibody with blocking peptide (5-fold excess, Santa Cruz Biotechnology Inc., Santa Cruz, CA, USA) for 2 h before being used for staining. Slides were mounted in DAKO mounting medium spiked with 5 μM Hoechst 33258 and viewed using Carl Zeiss Axiovert 200 M microscope.

### 2.7. Transmission Electron Microscopy

NT2/D1 and NT2-N/A (neurons were separately collected from astrocytes) were collected and fixed in 1.6% glutaraldehyde (*v/v*) in 100 mM sodium phosphate, pH 7.4 for 1 h at room temperature. Following washes with 100 mM sodium cacodylate, pH 7.2, cells were resuspended in 22% bovine serum albumin and pelleted by centrifugation at 900× *g* for 10 min. Pellets were then cut into 1 mm pieces, post-fixed with 1% osmium tetroxide for 1 h at 4 °C, stained en bloc with 2% uranyl acetate, dehydrated through graded concentrations of ethanol and embedded in Spurr's epoxy resin. Ultrathin sections were cut and stained with lead citrate. Photomicrographs were acquired on a Hitachi 7100 transmission electron microscope (TEM).

### 2.8. Immunoelectron Microscopy

Cells were fixed with 2% paraformaldehyde and 0.5% glutaraldehyde in 100 mM sodium phosphate buffer, pH 7.4 for 2 h at 4 °C. Following washes with 100 mM sodium cacodylate buffer, pH 7.2, cell pellets were prepared as described above, dehydrated through graded concentrations of ethanol and embedded in LRWhite resin at 4 °C. After polymerization in an oven at 50 °C, ultrathin sections were cut, placed on nickel grids coated with Formvar, and then processed for immunocytochemistry. Sections were first blocked with 1% BSA in PBS for 30 min and then floated on anti-caveolin-1 antibody (1:50) diluted in 1% BSA. After rinsing with PBS, sections were floated again on drops of secondary goat anti-rabbit IgG 15 nm-gold-conjugated antibodies (1:100, EY Laboratories, San Mateo, CA, USA). Incubation with primary antibodies was carried out at 4 °C for 24 h

and secondary antibodies at RT for 1 h, respectively. Sections were rinsed with distilled water, air dried and then counterstained with uranyl acetate and lead citrate. Negative controls consisted of omission of the primary antibodies. Photomicrographs were captured on a Hitachi 7100 TEM.

*2.9. Measurement of Endocytosis of Cholera Toxin (CTX-B)*

Fluorescent staining of lipid raft domains was performed using cholera toxin patching method, as described previously [35]. Cells were first incubated with 2 μg/mL cholera toxin B-subunit conjugated to Alexa 488 (CTX-B, Sigma) in HBSS containing 0.1% BSA for 30 min on ice. To cross-link lipid rafts (patching), cells were subsequently incubated with anti-CTX-B antibody (1:250, Calbiochem, San Diego, CA, USA) for 30 min on ice, and then for 20 min at 37 °C. Cells were fixed with Genofix for 10 min and immunostained for caveolin-1, as described above. Images were captured using Axiovert 200 M microscope (Carl Zeiss).

NT2/D1 or NT2/A cells were pretreated at 37 °C for 20 min with 0.5 mM methyl-β-cyclodextrin (mβCD), 20 μg/mL nystatin (Nys), 20 μg/mL filipin (Fil) or 50 μg/mL chlorpromazine (CP) (all reagents from Sigma). Cells were then washed with HBSS and incubated at 37 °C with 2 μg/mL CTX-B for 30 min. CTX-B uptake was analyzed using a Coulter ELITE ESP flow cytometer or CytoFluor^TM 2350 plate reader (Millipore, Bedford, MA, USA). NT2/A and NT2/N were grown as pure cultures, treated as above and analyzed using CytoFluor^TM 2350 plate reader. In a separate experiment, NT2/D1 and NT2/A cells grown on glass coverslips were processed as above, fixed with Genofix for 10 min and immunostained using caveolin-1 antibody.

*2.10. Statistical Analysis*

NT2 cultures were analyzed in triplicate and each experiment was repeated at least three times. All data are expressed as mean $\pm$ SD. Differences among means for all experiments were analyzed by using one-way ANOVA followed by a post hoc Dunnett's test using GraphPad Prism.

**3. Results**

*3.1. NT2 Precursor Cells Terminally Differentiate into Post-Mitotic NT2/N Neurons and NT2/A Astrocytes*

Human NT2/D1 precursor cells grew rapidly, with a doubling time of $25 \pm 4.2$ h, and morphologically resembled neuroepithelial cells [31]. A 4-week application of retinoic acid resulted in a terminally differentiated mixed population of NT2/N neuron-like cells and NT2/A astrocyte-like cells [32,33,36]. It has been previously established by Pleasure et al., that only $4.4 \pm 0.8\%$ of the initial population becomes terminally differentiated into NT2/N neurons [30]. After 3–4 weeks of growth in the culture, NT2/N was established as a network of interconnected neurons and were found resting on top of a monolayer of NT2/A that morphologically resembled "type I or protoplasmic" astrocytes (Figure 1A,C). NT2/N neurons did not incorporate bromodeoxyuridine (not shown) or show immunoreactivity to the anti-Ki-67 antibody [37], a widely used proliferation marker (Figure 1D); only $6.2 \pm 1.0\%$ of astrocytes were labeled with Ki-67 after 4 weeks of growth in the culture. In the mixed cultures, NT2/N expressed a microtubule associated protein-2 (MAP-2), a neuron-specific filament, and NT2/A expressed a glial fibrilliary acidic protein (GFAP), a glial-specific intermediate filament (Figure 1B).

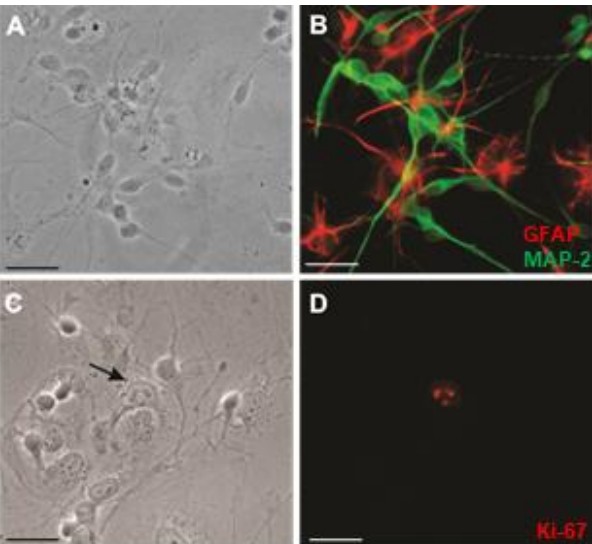

**Figure 1.** Immunofluorescence labeling of mixed NT2-N/A cultures. (**A**,**B**) Cultures were co-labeled with anti-MAP2 (green) and anti-glial fibrillary acidic protein (red) antibodies as described in Materials and Methods. In B, Neurons (green) grew on top of a monolayer of flat astrocytes (red); A, shows the phase-contrast image. (**C**,**D**) Cultures were labeled with anti-Ki-67 (red) antibody, as described in Materials and Methods. In D, a single proliferating astrocyte was identified by Ki-67; C, shows the phase-contrast image. Representative images are shown from at least three cultures. Magnification: ×400; Scale bars = 20 μm.

### 3.2. Caveolin Proteins Are Differentially Expressed in NT2 Cells

One of the major signature proteins of caveolae are caveolins [7]. Since caveolins are generally induced during the differentiation process, we first examined the expression of caveolins in NT2 cells by RT-PCR analysis (Figure 2). Caveolin-1 mRNA was dramatically upregulated in NT2/A astrocytes, compared to NT2/D1 precursors; however, it was downregulated in NT2/N neurons (Figure 2A, top panel). NT2/A astrocytes, allowed to mature in the culture for 6 weeks, showed a further increase in caveolin-1 mRNA, compared to NT2/A grown in the culture for 3 weeks. Caveolin-2 is a homolog of caveolin-1 and the two proteins are thought to form hetero-oligomers in most differentiated cell types [38]. We next determined whether the upregulation of caveolin-1 was also accompanied by upregulation of caveolin-2 in NT2/A. Consistently, there was also a marked increase in the expression of caveolin-2 mRNA in NT2/A astrocytes (Figure 2A, middle panel). In contrast to the lack of expression of caveolin-1 mRNA in NT2/N, caveolin-2 was detected in these cells. On the other hand, caveolin-3, known to be expressed mainly in striated and cardiac myocytes [38], was found to be expressed at similar levels in all NT2 cells (Figure 2A, bottom panel). These results demonstrate a differential expression of caveolin-1 and -2 in proliferating NT2/D1 versus differentiated NT2/A and NT2/N.

RT-PCR analysis using caveolin-2 specific primers revealed a PCR product of the expected size of 451 bp and a lower band of 268 bp. To confirm the expression of this 268 bp PCR product, we performed RT-PCR on several different cultured brain cells and a human epidermoid carcinoma cell line, A431 (Figure 2B). Consistent with an increased expression of the 451 bp product of caveolin-2 in NT2/A astrocytes, an increased expression of this mRNA was also observed in the terminally differentiated primary cultures of human brain endothelial (HBEC) and human fetal astrocytes (HFAS), compared with the tumorigenic human U87MG glioblastoma, and NT2/D1 cells. Caveolin-2 mRNA was expressed at the lowest level in the human A431 cells. Similar to the expected 451 bp PCR product, the novel 268 bp product was also expressed, not only in the NT2-derived cell types, but also in the primary cultures of HBEC and HFAS. As evident from Figure 2B, the expression level of the 268 bp mRNA was lower in U87MG and the lowest in A431 cells. It should be noted that although the 268 bp product was readily detectable with 30 cycles of PCR amplification,

the expression level was obviously lower than that of the full length 451 bp mRNA. The sequence of the 268 bp PCR product was determined after subcloning it into a plasmid vector with the subsequent BLAST analysis. The analysis revealed a deletion in exon-2 which resulted in a 1314 bp mRNA, compared to the 3332 bp mRNA encoded by the full length caveolin-2 gene (GenBank accession no. NM_001233). The nucleotide sequence data of our novel splice variant was registered in the nucleotide sequence database (GenBank accession no. AY353255). Figure 2C shows the schematic representation of the caveolin-2 gene with its known mRNA transcript and our novel splice variant.

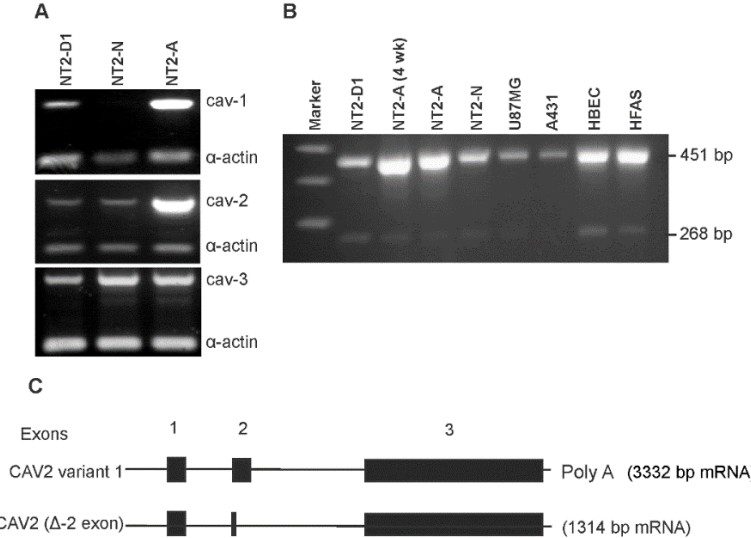

**Figure 2.** Expression of caveolins in NT2 cells. Total RNA was extracted and RT-PCR analysis of caveolin mRNAs was carried out using primers specific for each caveolin isoform (Table I) as described in Materials and Methods. (**A**) RT-PCR amplification of caveolins in NT2 cells. Top panel shows the expression levels of caveolin-1, middle panel shows the expression levels of caveolin-2 and bottom panel shows the expression levels of caveolin-3. α-actin was used as a control gene. (**B**) RT-PCR amplification of caveolin-2 splice variant in various cell types. The full length mRNA was detected as a 451 bp PCR product, while the novel caveolin-2 splice variant was detected as a 268 bp PCR product. Representative gels are shown from at least four independent experiments. (**C**) Schematic representation of human caveolin-2 gene and its full length mRNA, and the novel caveolin-2 splice variant. Exons are numbered and indicated by boxes; introns are represented by horizontal lines. Patterns of alternative splicing of the splice variant are shown.

The subcellular distribution of caveolin-1 was evaluated by indirect immunofluorescence in NT2 cultures. As shown in Figure 3, caveolin-1 was expressed in NT2/D1 and NT2/A cells, however, their patterns of expression were considerably different. NT2/A astrocytes, grown either as mixed (Figure 3B) or pure (Figure 3C,D) cultures, showed a punctuate caveolin-1 immunoreactivity distributed throughout the cell, decorating the plasmalemmal surfaces, and particularly enriched in the astrocytic processes (Figure 3D). Furthermore, caveolin-1 was found to be concentrated in puncta of various sizes, located around the perinuclear and Golgi region (Figure 3B,D). The glial identity of the cells was confirmed by double immunolabeling carried out on the differentiated NT2/A cultures with caveolin-1 and glial fibrilliary acidic protein, a marker for astroglial cells (Figure 3C). On the other hand, NT2/D1 cells showed diffused caveolin-1 immunoreactivity scattered throughout the cells and no caveolin-1 immunoreactivity was found to be concentrated into puncta or micropatches on the plasma membrane. Immunofluorescence was completely eliminated in cultures stained with the caveolin-1 antibody, preincubated with the blocking peptide. The expression of caveolin-3 was also examined by immunocytochemistry. While caveolin-3 was expressed uniformly throughout the cytoplasm of NT2/D1 cells (not shown), NT2/N and NT/A cells (Figure 3E,F) showed distinct expression patterns,

suggesting that caveolin-3 may be expressed in different compartments of the cell. In NT2/N, caveolin-3 immunoreactivity was detected throughout the soma, including high levels in growth cones (Figure 3E). NT2/A cells showed uniform diffuse caveolin-3 immunoreactivity distributed throughout the cell, with very few dense micropatches on the plasma membrane. A similar pattern of caveolin-3 staining was seen in the primary mouse cortical and hippocampal neurons (not shown).

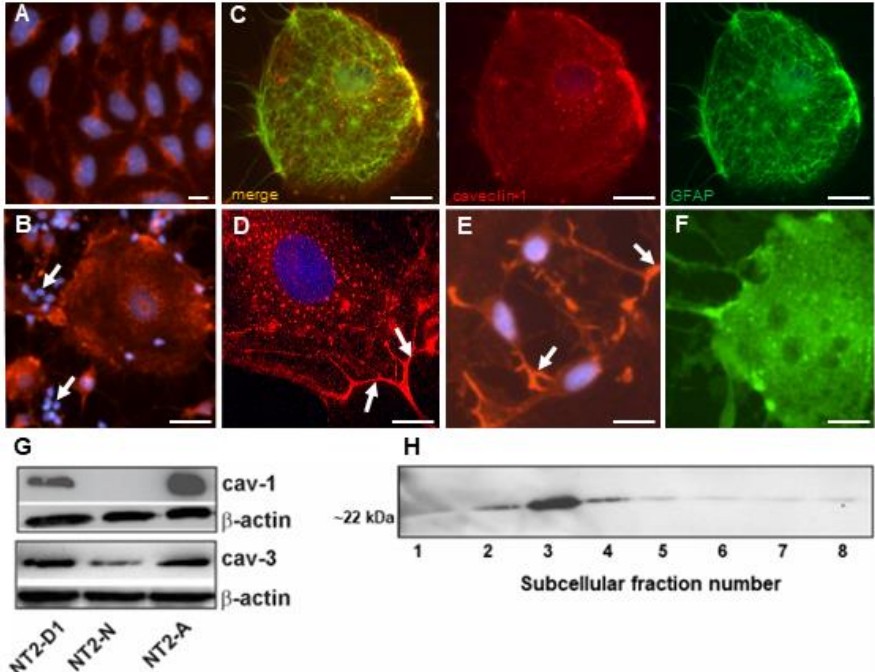

**Figure 3.** Expression of caveolin-1 and caveolin-3 in NT2 cells. (**A**–**D**) Cells were grown, fixed and processed for immunocytochemistry using anti-caveolin-1 antibody, as described in Materials and Methods. (**A**) NT2/D1 cells showed a diffused caveolin-1 immunoreactivity distributed throughout the cell. (**B**) NT2/A in mixed NT2-N/A cultures showed caveolin-1 immunoreactivity organized into small fluorescent puncta that are scattered throughout the cell, and larger clusters that appeared concentrated in the centrosomal region. At the plasma membrane, caveolin-1 immunoreactivity was also concentrated into small and large micropatches. Arrows indicate absence of caveolin-1 immunoreactivity in NT2/N neurons. (**C**) Pure NT2/A cells were double labeled with anti-glial fibrillary acidic protein and anti-caveolin-1 antibodies, respectively. (**D**) Pure NT2/A cells showed similar caveolin immunoreactivity as NT2/A in mixed cultures. Arrows indicate astrocyte processes. (**E**) NT2/N in mixed NT2-N/A cultures were labeled with anti-caveolin-3 antibody, as described in Materials and Methods. NT2/N showed diffuse caveolin-3 immunoreactivity throughout the soma, which was mainly concentrated in growth cones (arrows). (**F**) NT2/A cells showed uniform diffuse caveolin-3 immunoreactivity distributed throughout the cell with very few dense micropatches on the plasma membrane. Nuclei (blue) were stained with Hoechst 33258. Representative images are shown from at least four independent experiments. Scale bars = 10 μm. (**G**) Western blot analysis of caveolins in NT2 cultures. Protein extracts were prepared from various NT2 cell types, subjected to SDS-PAGE and probed with anti-caveolin-1 or -3 antibodies, respectively. β-actin was used as a loading control. (**H**) Western blot analysis of caveolin-1 in subcellular fractions of NT2/A. Subcellular fractions from NT2/A cells were prepared as described in Materials and Methods. Equal amounts of protein (5 μg) from each fraction was resolved by SDS-PAGE and processed for immunoblot analysis using anti-caveolin-1 antibody. Fraction number is indicated below the immunoblot. Representative gels are shown from at least two experiments.

Western blot analysis of protein lysates revealed a single immunoreactive band of caveolin-1 at ~22 kDa, which was upregulated by ~1.5-fold in NT2/A astrocytes, compared

to NT2/D1 precursors and being undetectable in NT2/N (Figure 3G). On the other hand, a single immunoreactive band of caveolin-3 at ~20 kDa (comparable to the size of muscle caveolin-3), was expressed at similar levels in undifferentiated and differentiated NT2 cells (Figure 3G). To determine if caveolin-1 was found in caveolae, NT2/A astrocytes were subjected to subcellular fractionation under detergent-free conditions to isolate caveolae [34]. As shown in Figure 3H, the light-density fractions 2, 3 and 4 showed the highest caveolin-1 immunoreactivity, as previously demonstrated in other cell types, such as primary cultures of rat type I astrocytes, C6 glioma and human glioblastoma cells [13,14,19]. Taken together, these data strongly support the notion that the expression of caveolins can be regulated by retinoic acid, and caveolin-1 and -2 were induced in terminally differentiated NT2/A astrocytes but not in NT2/N neurons.

### 3.3. Caveolae-like Structures Are Abundantly Expressed in Differentiated NT2/A and NT2/N Cells

In order to study the distribution of caveolae, various NT2 cultures were processed and subjected to TEM. As shown in Figure 4, caveolae were easily distinguishable from the clathrin-coated pits by their small size, shape and lack of the clathrin basket. Clathrin-coated pits and vesicles generally appeared as ~250–300 nm electron-dense invaginations of the plasma membrane (Figure 4A,B,D), which were observed in all the NT2 cell types examined. Caveolae appeared as small pear-, omega- or flask-shaped invaginations of the plasma membrane, which were detected at an abundant level in NT2/A astrocytes, grown as pure (Figure 4A) or mixed cultures (Figure 4B). Caveolae-like structures opening to the outside milieu were found decorating the plasma membrane of NT2/N neurons (Figure 4C), and a large number of free vesicular structures were also seen immediately below the plasma membrane and in the cytoplasm. Additionally, clusters of interconnecting caveolae forming rosettes were also seen in the NT2/A cultures (Figure 4A,B, insets). In contrast, caveolae-like structures were expressed at very low numbers in NT2/D1 precursors (Figure 4D). Caveolae size was determined by measuring their diameter from the TEM photomicrographs (Table 2). All cell types expressed caveolae-like structures at a remarkably constant diameter of between 80 and 100 nm.

**Table 2.** Expression of caveolae-like structures in NT2 cells.

| Cell Type | Size (nm) | Number of Caveolae-like Structures Measured |
|-----------|-----------|---------------------------------------------|
| NT2/A (4 wk) | 97.6 ± 16.7 | 40 |
| NT2/A | 94.8 ± 11.5 | 34 |
| NT2/N | 102 ± 7.7 | 61 |
| NT2/D1 | 86 ± 20.8 | 12 |

NT2/D1, pure NT2/A (4 wk old) and NT2/A and NT2/N (collected from mixed NT2-N/A cultures) were embedded for electron microscopy and analyzed as described in Materials and Methods. Data are shown as mean ± S.D.

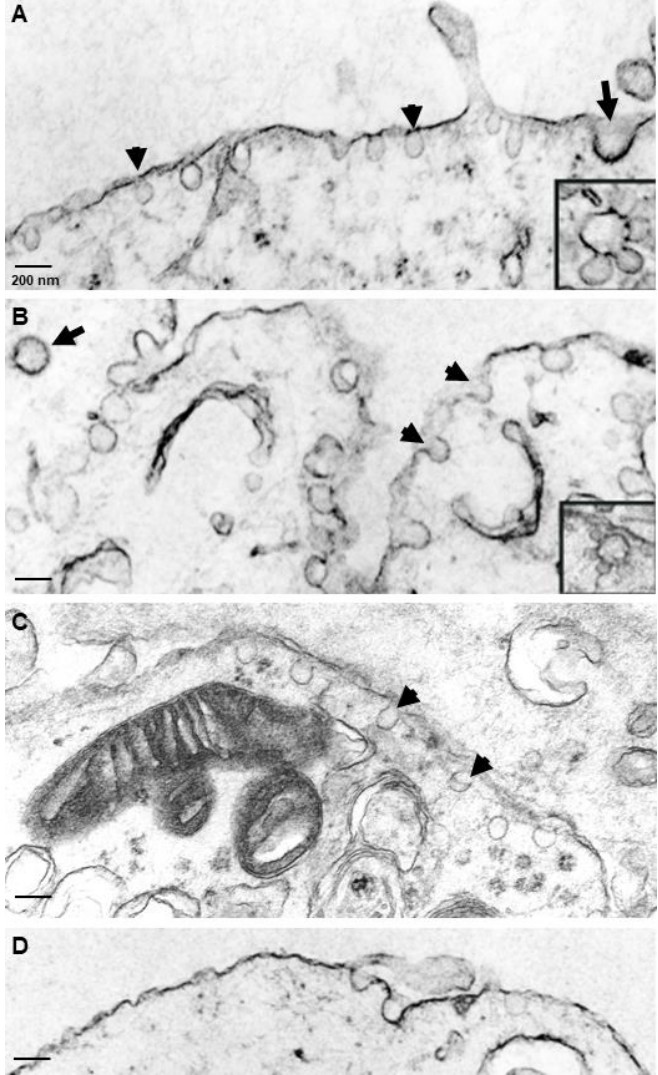

**Figure 4.** Ultrastructural localization of caveolae in NT2 cells. Cells were fixed in glutaraldehyde and processed for electron microscopy, as described in Materials and Methods. NT2/A, grown as pure (**A**) or mixed cultures (**B**), showed abundant omega- or flask-shaped invaginations of the plasma membrane, ~100 nm in diameter, which were open to the extracellular milieu or found as free vesicles beneath the plasma membrane. Insets shows clusters of vesicles forming rosettes. NT2/N from mixed cultures (**C**) also showed omega-shaped invaginations present on the plasma membrane and abundant free vesicles beneath the plasma membrane. Arrow shows a clathrin-coated pit in each cell type. Arrowheads show caveolar structures. NT2/D1 showed only a few plasma membrane invaginations (**D**). Scale bars = 200 nm.

### 3.4. Caveolin-1 Is Localized to Caveolae in NT2/A Astrocytes

To further determine whether the signature protein of the caveolae, caveolin-1, indeed localizes to the caveolar structures in NT2/A astrocytes, we used immunoelectron microscopy (Figure 5). Immunogold particles, corresponding to the binding sites of anti-caveolin-1, were found decorating the caveolae on the plasma membrane and small vesicles in the cytoplasm (Figure 5A,B), lending further evidence that the fluorescent micropatches of caveolin-1 immunoreactivity correspond to collections of caveolae. Numerous caveolin-1 antigenic sites were also seen on the length of the processes of the astrocytes (Figure 3D). No gold labeling was seen in sister specimens incubated with a non-immune IgG.

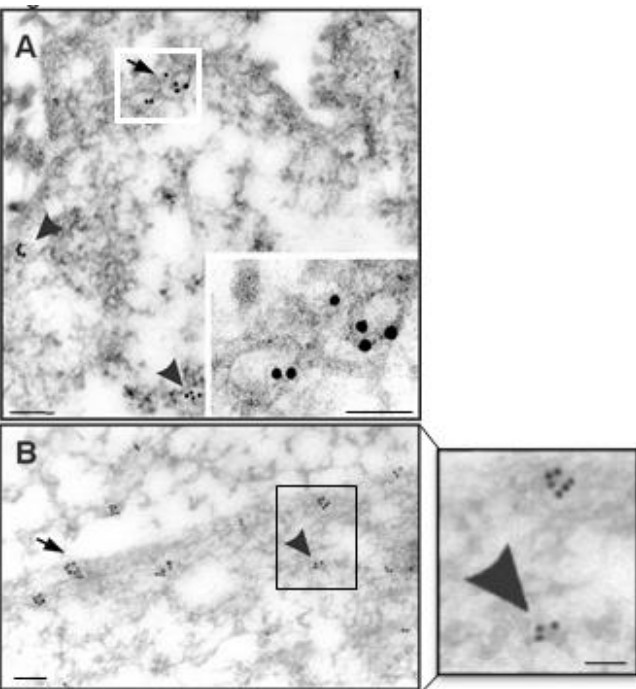

**Figure 5.** Immunoelectron localization of caveolin-1 in NT2/A. Cells were fixed in a mixture of paraformaldehyde/glutaraldehyde and processed for immunoelectron microscopy, as described in Materials and Methods. NT2/A, grown as pure (**A**) or mixed cultures (**B**), showed abundant 15-nm gold particles corresponding to caveolin-1 immunoreactivity in the caveolar structures on the plasma membrane (arrow) and as free vesicles in the cytoplasm (arrow head). Scale bars in A and B = 200 nm; inset in A = 300 nm and inset in B = 200 nm.

### 3.5. Cholera Toxin B-Subunit (CTX-B) Colocalizes with Caveolin-1 in Lipid Rafts

Lipid rafts were visualized using Alexa 488 conjugated cholera toxin B-subunit (CTX-B), which binds the ganglioside GM1 in lipid rafts and caveolae [39], and then cross-linked into patches with an anti-CTX-B antibody [4]. Figure 6A shows that these domains were densely distributed throughout the plasma membrane of NT2/A astrocytes. In cells patched with the anti-CTX-B antibody, lipid rafts aggregated into patches and CTX-B appeared to be clustered in these discrete patches, as evident in Figure 6B. To determine whether CTX-B co-localized with caveolin-1, immunocytochemistry was performed. Caveolin-1 staining overlapped with CTX-B in the plasma membrane (Figure 6C). It has been shown in other cell types that cholesterol depletion of cell membranes leads to the loss of caveolar structures [6,40]. Exposure of NT2/A cells to mβCD, which extracts cholesterol from the plasma membrane, resulted in a discontinuous, patchy membrane staining of CTX-B, and caveolin-1 staining clustered into patches in the cholesterol and caveolae-depleted membrane (Figure 6D). Similar results were obtained after treatment of NT2/A with nystatin and filipin, drugs that bind cholesterol in the plasma membrane and impair the invagination of caveolae, thereby inhibiting caveolae internalization.

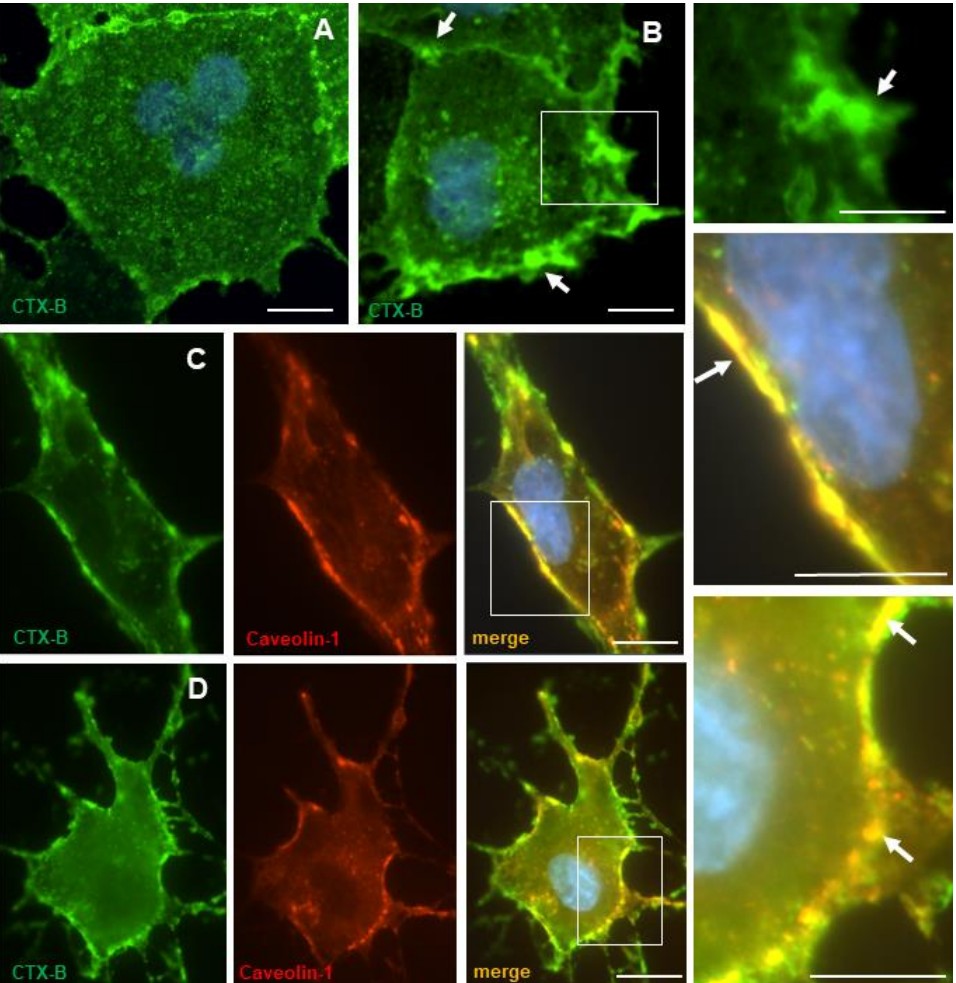

**Figure 6.** Localization of lipid rafts in NT2/A. Cells were incubated with CTX-B and left untreated or patched with anti-CTX-B antibody. Following fixation, CTX-B was visualized using fluorescence microscopy, as described in Materials and Methods. (**A**) In untreated control cells, CTX-B appeared to be uniformly distributed at the plasma membrane. (**B**) In patched cells, CTX-B appeared to be concentrated in distinct patches (arrow) at the plasma membrane. White box in B has been enlarged to show CTX-B clustered on the plasma membrane. (**C,D**) Cells incubated with CTX-B in the absence or presence of mβCD were fixed and then processed for immunocytochemistry using anti-caveolin-1 antibody. (**C**) In untreated control cells, caveolin-1 immunoreactivity colocalized with CTX-B at the plasma membrane. (**D**) In cells treated with mβCD, caveolin-1 immunoreactivity appeared as a discontinuous patchy membrane staining. White boxes in C and D have been enlarged to show colocalization. Nuclei (blue) were stained with Hoechst 33258. Scale bars = 10 μm.

*3.6. Cholera Toxin B-Subunit (CTX-B) Is Internalized by Different Mechanisms in NT2/A and NT2/N Cultures*

To determine the functionality of caveolae in NT2 cultures, we tested the internalization of CTX-B, a bacterial protein, which has been considered to be a good marker for caveolae-mediated endocytosis [41]. NT2 cells were incubated in the presence of CTX-B at 4 °C for 30 min, and then internalization was allowed to proceed at 37 °C for 30 min in the absence or presence of selective inhibitors of caveolae or clathrin-dependent endocytosis. Figure 7 shows the internalization of CTX-B in NT2/D1, NT2/A and NT2/N cells. After incubation for 30 min at 4 °C, labeling was mainly associated with cell surface in NT2/A and some labeling in NT2/N cells, but not much labeling was found on NT2/D1 cells. However, after 30 min of incubation of NT2/A cells at 37 °C, we observed that endocytosed CTX-B appeared as bright fluorescent puncta underneath the plasma membrane and

scattered throughout the cytoplasm. The labeling was explicitly seen in the perinuclear and Golgi region, although some labeling remained at the cell surface. A significant proportion of the endocytosed CTX-B was found to colocalize with caveolin-1-positive endosomes (Figure 7A, panel e, merge). On the other hand, in NT2/N, CTX-B appeared as a bright fluorescent signal, associated with the plasma membrane, and there were very few fluorescent puncta in the cytoplasm (Figure 7B, panel b). In NT2/D1, no significant amount of endocytosed CTX-B was evident (Figure 7A, panel b, merge) and no colabeling was found in the caveolin-1-positive endosomes.

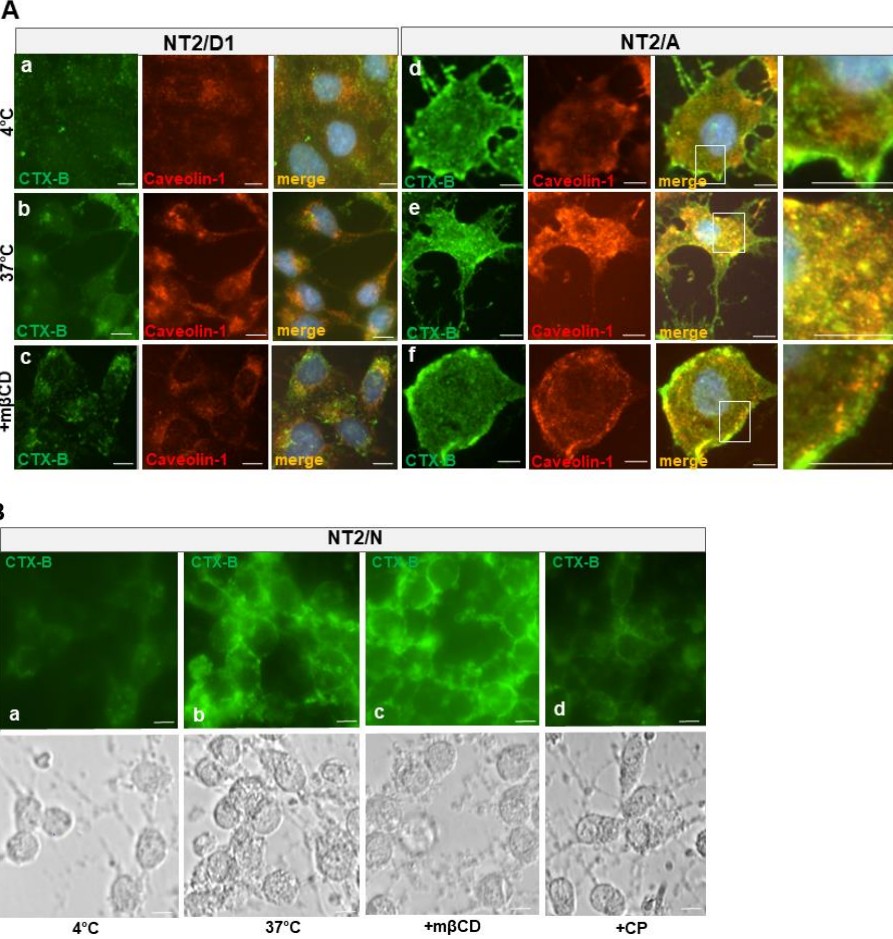

**Figure 7.** Subcellular localization of CTX-B internalized by NT2 cells. (**A**) NT2/D1 (top panels) or NT2/A (bottom panels) cells were incubated with CTX-B at 4 °C for 30 min (panel a, d), fixed, and processed for immunocytochemistry using anti-caveolin-1 antibody. CTX-B labelling (green) appears on the plasma membrane in NT2/A and colocalizes with caveolin-1 immunoreactivity (red) in the lipid rafts (panel d). The merged images depicting colocalization (yellow) of CTX-B and caveolin-1 immunoreactivity are shown. Incubation of NT2/A cells (panel e) with CTX-B at 37 °C for 30 min resulted in the internalization of CTX-B and abundant colocalization with caveolin-1 in the cytoplasm. By contrast, incubation of NT2/D1 (panel b) with CTX-B at 37 °C for 30 min did not show significant uptake of CTX-B. Pretreatment of cells with mβCD (panel c and f), followed by incubation with CTX-B, did not result in endocytosis of CTX-B, most remained entrapped in the plasma membrane. White boxes in panels d, e and f have been enlarged to show CTX-B at the plasma membrane (d and f) or colocalizing with caveolin-1 (e). (**B**) NT2/N cells (panel a) were incubated with CTX-B at 4 °C for 30 min or at 37 °C for 30 min (panel b). In some experiments, cells were pretreated with mβCD or CP and then incubated with CTX-B at 37 °C for 30 min (panels c and d). Shown are also phase contrast images of pure cultures of NT2/N. Images shown are representative of at least two experiments with similar results. Scale bars = 10 μm.

NT2 cells were treated with several cholesterol binding agents such as filipin, nystatin or mβCD, which modify the structure and function of caveolae [42] and chlorpromazine (CP), which caused an aberrant endosomal accumulation of clathrin. For example, treatment of NT2/A with mβCD resulted in the disruption of caveolae-dependent endocytosis of CTX-B. Accordingly, CTX-B remained mainly entrapped at the plasma membrane, with a patchy colocalization with caveolin-1 (Figure 7A, panel f, merge). Some CTX-B was seen as small puncta scattered throughout the cytoplasm, with little colocalization with caveolin-1 (Figure 7A, panel f, merge). Pure cultures of NT2/N were also treated with inhibitors of caveolae and clathrin-dependent endocytosis (Figure 7B). In contrast to the NT2/A cells, mβCD had no effect on the CTX-B internalization in NT2/N, while CP resulted in a dramatic reduction of CTX-B internalization.

The amount of CTX-B endocytosed and the effect of cholesterol depleting agents in various NT2 cell types was also quantitated by fluorescence-based assays (Figure 8). Incubation of NT2/D1 cells at 37 °C for 30 min with CTX-B did not result in a significant internalization of CTX-B; no effect was observed with the inhibitors of either the caveolae- or clathrin-pathway. On the other hand, NT2/A astrocytes and NT2/N neurons resulted in the accumulation of CTX-B on the plasma membrane and subsequent internalization; however, this uptake was mediated by separate pathways. In NT2/A astrocytes, CTX-B internalization was inhibited to a similar level by the inhibitors of the caveolin- and clathrin-pathway (Figure 8A,B). By contrast, in NT2/N neurons, mβCD and filipin had no effect on the rate of CTX-B internalization while treatment with CP resulted in a statistically significant reduction of CTX-B internalization (Figure 8C).

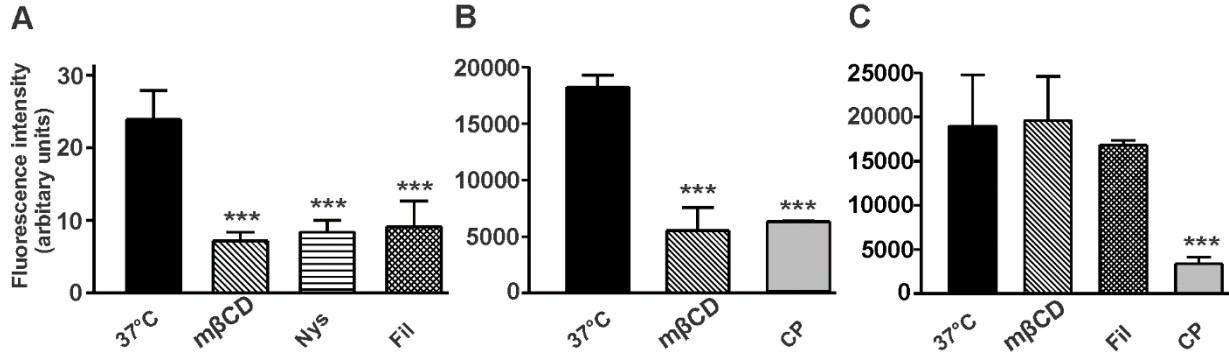

**Figure 8.** Analysis of CTX-B internalization in NT2 cells. Cells were treated in the presence or absence of inhibitors of the caveolin- or clathrin-pathway, and internalization of CTX-B was allowed at 37 °C for 30 min. NT2/A from mixed cultures were collected by trypsinization and analyzed by flow cytometry (**A**). NT2/A (**B**) and NT2/N (**C**) were grown as pure cultures and analyzed by fluorescence plate reader, as described in Materials and Methods. Internalization of CTX-B at 4 °C has been subtracted from all samples. Data is shown as mean ± SD from a representative experiment performed three independent times. Statistically significant differences are shown as *** ($p < 0.001$).

## 4. Discussion

The presence and role of caveolae and caveolins has been widely studied in various non-neural cells and recently their role is under investigation in neural cells. In this study, by using various morphological and biochemical methods, we have shown for the first time that retinoic acid treatment of NT2/D1 progenitor cells was associated with a remarkable alteration of the plasma membrane in the differentiated NT2/N neuron-like cells and NT2/A astrocyte-like cells, as evidenced by electron microscopy and changes in the expression of the caveolin proteins (Figures 3 and 4).

We found a differential expression of caveolin proteins in NT2 cultures, and, in particular, caveolin-1 and -2 mRNA and protein levels were upregulated during the differentiation of NT2/D1 progenitors into NT2/A astrocyte cultures. Consistent with this observation, both caveolae and caveolins have been shown to be most abundantly expressed in terminally differentiated cells [43]. For instance, undifferentiated human keratinocytes had an

undetectable expression of caveolin-1 and -2 at the mRNA and protein level; however, these messages were induced upon differentiation with the keratinocyte growth factor [44]. Similarly, a 9-fold increase in the caveolar invaginations and caveolin-1 expression was seen during differentiation of mouse 3T3-L1 fibroblasts into adipocytes, whereas no change in the clathrin invaginations was evident [45]. Furthermore, increased levels of caveolin-1 and -2 protein expression was seen in rat pheochromocytoma PC-12 cells, differentiated with the nerve growth factor into neuron-like cells [17]. Robust expression of caveolin-1 and -2 was also observed along the entire cell surface of dorsal root ganglion neurons, including high levels on growth cones [17]. Interestingly, recombinant overexpression of caveolin-1 arrested mouse embryonic fibroblasts in the G0/G1 phase of the cell cycle, indicating a role for caveolin-1 in mediating growth arrest [46]. Therefore, the signaling pathways that lead to the expression of caveolin-1 and -2 in NT2/A astrocytes might be linked to the cell cycle. Indeed, a p53/p21-dependent pathway has been implicated in mediating G0/G1 growth arrest in mouse embryonic fibroblasts [46]. The expression of caveolin-1 was upregulated during the C6 glioma cells' differentiation, primary astrocyte cultures and human glioblastoma cells [13,47]. In neural progenitor cells, caveolin-1 promoted astrocyte differentiation through the modulation of Notch-1/NICD and the hairy enhancer of split-1 (Hes1) signaling [48]. Moreover, an increase in caveolin-1 expression was detected in NT2/A astrocytes that were allowed to mature in the culture, further strengthening the link between caveolins, the cell-cycle exit and terminal differentiation. Taken together, caveolin-1 may serve to be a useful marker of astroglial maturation during gliogenesis. In this study, we have identified a novel splice variant of human caveolin-2 mRNA in the tested brain cells. This splice variant harbors a deletion in exon-2, resulting in the production of a 1314 bp mRNA, compared to the 3332 bp mRNA encoded by the full length caveolin-2 gene. A previous study has also documented a splice variant of mouse caveolin-2 mRNA, which localized in the endoplasmic reticulum [49]. The function of the human and the mouse caveolin-2 splice variants is not known. It will be interesting to identify the proteins interacting with these caveolin-2 splice variants, which might reveal novel function.

Although a great deal of attention has been focused on studying the intracellular signal transduction mediated by caveolin proteins, the molecular events leading to the formation of caveolae have not been carefully examined. A lack of caveolin-1 expression correlated with the absence of caveolae in a lymphocyte cell line [4] and mouse neuroblastoma N2a cells [50]. Consistent with its structural role, caveolin-1 expression directly correlated with the morphological appearance of caveolae [4]. In most eukaryotic cells, caveolin-1 is usually localized in the caveolae on the plasma membrane. NT2/D1 cells, which expressed a very low number of caveolae on the plasma membrane (Figure 4D), demonstrated a diffuse cytoplasmic caveolin-1 staining (Figure 3A), suggesting that it might be targeted to a different compartment of the cell. It is plausible that caveolin-1 might be present in the Golgi, consistent with its role in cholesterol transport and signal transduction. Indeed in cells expressing low levels of caveolin, labeling was dispersed over the plasma membrane without the formation of caveolae-like pits [4]. However, caveolins can also be targeted to a variety of intracellular destinations. In skeletal muscle and skin keratinocytes, caveolin-1 was targeted to the cytosol, while in exocrine and endocrine cells, it accumulated in the secretory pathway. We have also seen caveolin-1 immunogold particles localized to mitochondria in NT2/A (unpublished observations), as demonstrated previously in airway epithelial cells [51]. These authors further confirmed the specificity of the mitochondrial signal in various cell types by using various biochemical and molecular techniques.

Retinoic acid (RA) is a known regulator of stem cell differentiation [52]. In our studies, RA treatment induced differentiation of NT2/D1 progenitors into NT2/N neuron-like cells which expressed classical neuronal morphology and produced neuron-specific markers, MAP-2 and β-III tubulin, respectively (Figure 1) [33]. Caveolin-1 was selectively downregulated in NT2/N neuron-like cells (Figure 3). Consistent with our findings, Han et al., have recently demonstrated that downregulation of caveolin-1 in human adipose-derived mes-

enchymal stem cells was accompanied with an increase in the expression of mesencephalon-specific transcription factors, Lmx1a, Nurr1 and tyrosine hydroxylase in dopaminergic-like neurons [27]. Hence, caveolin-1 appears to be a negative regulator of neuronal cell differentiation. On the other hand, NT2/N cells expressed abundant amounts of caveolin-3, and electron photomicrographs revealed numerous omega-shaped caveolae-like vesicles on the plasma membrane (Figure 4). These data support the notion that, in NT2/N cells, caveolin-3 likely substitutes for caveolin-1 to drive caveolae formation. Previously, it has been demonstrated that the recombinant expression of caveolin-3 was sufficient to drive the formation of invaginated caveolae in muscle cells and Sf21 insect cells [53,54]. Shikanai et al., have shown that caveolin-1 is predominantly expressed in immature cortical neurons and regulates the early phase of neuronal maturation via caveolin-1-mediated trafficking and endocytosis of N-cadherin and L1 cell adhesion molecules [21]. Using differentiated human neuronal progenitor cells, derived from induced pluripotent stem cells (iPSCs), a recent study has shown that caveolin-1 phosphorylation is a key regulator of axonal growth during the early stages of neuronal differentiation [55]. Evidence suggests that caveolin-1 has a complex role during neural stem cell differentiation and warrants further investigation.

There are numerous endocytosis pathways by which a variety of cargo molecules are internalized from the surface of eukaryotic cells [56], and the complexity of the endocytic machinery is well documented [57]. Our studies confirmed that NT2/N and NT2/A endocytosed and transported CTX-B to the cytoplasm (Figures 7 and 8). Accordingly, we also found that NT2/A endocytosed CTX-B by both the clathrin- and caveolae-dependent pathways, since the inhibitors of caveolae- and clathrin-pathway inhibited CTX-B uptake. NT2/N contained caveolae but no caveolin-1, and nystatin and filipin had no effect on CTX-B uptake, indicating major involvement of clathrin-coated pits in CTX-B internalization. Internalization of some caveolar ligands has been shown to be a signal mediated process that requires caveolin-1 expression. In many systems caveolae are thought to serve as communicators between the membrane bilayer and the cell cytoplasm. Additionally, it has been shown that a significant amount of CTX can enter various cell types through a clathrin-dependent pathway [42,58,59]. The route by which CTX-B enters the cell may also depend on the level of ganglioside GM1 expression at the plasma membrane [60]; when GM1 levels are increased, the pathway for CTX-B uptake becomes more sensitive to cholesterol depletion, independently of caveolin-1 expression [60]. The relationship between caveolin-1 expression and the choice of the endocytosis pathway is not known, although, in several studies, caveolin-1 expression has been shown to inhibit rather than stimulate clathrin-independent endocytosis [61,62].

Astrocytes are the most abundant glial cell in the brain that play a crucial role in regulating almost all homeostatic functions, including maintenance and regulation of the blood-brain barrier integrity. The presence of caveolins in astrocytes has been widely documented and the functional significance of this compartment under pathophysiological conditions is increasingly becoming clear [63]. During oxidative stress, caveolin-1 has been shown to act as a checkpoint regulator in astrocytes [64]. Exposure of primary astrocytes to hydrogen peroxide induced the expression of aquaporin-4 (AQP4), a water-permeable channel, which was shown to be regulated by caveolin-1 phosphorylation [65]. Under these conditions, caveolin-1 was shown to bind to Src homology 2-containing protein tyrosine phosphatase 2 (SHP-2) and function as a positive regulator of Src signaling, resulting in neuroprotection [66]. These effects were abrogated by incubation with N-acetylcysteine and the Src kinase inhibitor, PP2. Furthermore, loss of caveolin-1 in astrocytes resulted in a decrease in AQP4, a reactive astrogliosis and increased brain edema [67]. Astrocytes play an important role in supporting neuronal health, for example, astrocytes affect neurogenesis, neuronal survival, neurite outgrowth, and synapse formation [68,69]. However, it is still unknown whether caveolin-1 expression is essential to generate or maintain the differentiated state of astrocytes, as is its significance in neuron-astrocyte communication.

Accumulating evidence points to caveolae and caveolin proteins as critical factors in the pathophysiological processes in the brain. The aberrant expression of caveolin proteins may lead to impairments in signal transduction pathways, leading to the development of neurodegenerative diseases, such as Alzheimer's disease (AD), Parkinson's disease (PD), Huntington's disease (HD) and Amyotrophic lateral sclerosis (ALS) (reviewed by [70]). Earlier studies have documented that amyloid precursor protein (APP, a source of Aβ amyloid peptide) is enriched within caveolae in neurons [71] and colocalizes with caveolin-1 [72]. Accordingly, the overexpression of caveolin-1 promoted the β-secretase-mediated cleavage of APP, resulting in Aβ accumulation [72]. Similarly, the overexpression of caveolin-1 is also associated with dysregulation of cholesterol homeostasis in brain cells, which may lead to AD [73]. Caveolin-3 has also been reported to be upregulated along with presenilin-1 and presenilin-2 in reactive astrocytes surrounding amyloid plaques [74]. The recombinant overexpression of caveolin-3 promoted the β-secretase-mediated processing of APP, suggesting that reactive astrocytes may produce harmful metabolites of APP. Caveolin-1 has been shown to directly interact with mutant huntingtin (mHTT) protein and confer a "gain-of-function' that promotes neurotoxicity [75]. However, in the SOD1 mouse model of ALS, targeting caveolin-1 to motor neurons delayed disease progression, probably by augmenting neurotrophin signaling [76]. Several PD-related gene products, such as α-synuclein, LRRK2, PINK1, parkin and DJ-1, have been reported to associate with lipid rafts via directly interacting with caveolin-1 [70]. Alterations in lipid rafts by the loss of parkin was shown to disrupt the ubiquitination and degradation of caveolin-1, resulting in increased caveolin-1 in neurons and aberrant signaling [77]. DJ-1 deficiency modulated lipid raft-dependent endocytosis through flotilin-1 and caveolin-1 in astrocytes, resulting in impairment of glutamate clearance [78]. In summary, the increasing number of studies reveal a complex role of caveolins in neurodegenerative diseases, where they can exert both protective and destructive effects by regulating diverse pathways.

## 5. Conclusions

The role of caveolins in a variety of signaling pathways in neural cells is increasingly being appreciated; however, the relationship between these functions and endocytosis pathways in human neuron-astrocyte cultures remains understudied. Although human iPSCs can be differentiated into neurons and astrocytes, and have emerged as useful platforms for disease modeling, these cells require special media and are expensive. Therefore, mixed cultures of human NT2/N neurons and NT2/A astrocytes may prove to be a useful experimental model to study the role of caveolin proteins in subcellular transport and metabolism, as well as in deciphering neuron-astrocyte communication.

**Author Contributions:** J.K.S. and A.A. conceptualized the project and carried out the experimental work. They also collected and analyzed the data. J.K.S. prepared the figures and wrote the manuscript. M.R.-L. carried out the RT-PCR analysis and the experimental work to identify the novel cav-2 variant. A.A. and M.R.-L. contributed to writing the methods. All authors have read and agreed to the published version of the manuscript.

**Funding:** This research was supported by intramural funding from the National Research Council Canada.

**Institutional Review Board Statement:** Not applicable.

**Informed Consent Statement:** Not applicable.

**Data Availability Statement:** Not applicable.

**Acknowledgments:** We sincerely thank Caroline Sodja and Claudie Charlebois for their excellent technical assistance in the Western blot analysis and sequencing the caveolin-2 splice variant. We also thank P Roy Walker, Marianna Sikorska (Emeritus researchers) and Danica Stanimirovic for their guidance.

**Conflicts of Interest:** The authors declare no conflict of interest.

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
