# Peer review of "Molecular and Functional Characterization of Caveolae in Mixed Cultures of Human NT-2 Neurons and Astrocytes"

_2571-6980, doi:10.3390/neuroglia2010008_

Round 1
Reviewer 1 Report
In this manuscript, the authors investigated the presence of caveolae and caveolin proteins in human NT2 cells differentiated neuron-like(NT2/N) and astrocyte-like (NT2/A) cells. They found differential expression of caveolin proteins in NT2 cultures, particularly the mRNA and protein levels of caveolin1 and caveolin2 were upregulated in NT2/A cells. These finding indicate that different endocytosis mechanisms may be involved in astrocyte and neurons.
The study appears to use appropriate statistical tests and includes methods that would allow for other researchers to reproduce the work.
The mixed cultures of human NT2/N neurons and NT2/A astrocytes may prove to be useful experimental model to study the role of caveolin proteins in protein trafficking, especially for unfolded proteins involved in neurodegeneration. For example, it seems that astrocytes and neurons are using different machineries to uptake Tau proteins. This model developed in the paper will be a useful in that direction.
Overall this work is well done. The quality of the data, and the interpretation of this data meet the standard expected of this journal. I recommend that the paper be accepted pending confirmation from other reviewers that their concerns were addressed sufficiently.
Author Response
No changes were requested.
Reviewer 2 Report
The authors studied caveolin expression and localization, as well as caveolae formation in NTera-2 (NT2/D1) cells which are differentiated to neuronal cell types (astrocytes or neurons). Therefore RT-PCR, Western Blot analysis, immunostaining and light microscopy, electron microscopy and a functional cell based Cholera toxin B (CTxB) assay was used. The authors showed that astrocytes (NT2/A) contain strong caveolin expression and caveolae membrane domains at the plasma membrane. Additionally, they detected caveolae mediated CTxB endocytosis in astrocytes. Next, the authors analyzed neuron cultures (NT2/N), in which they detected caveolin 2 and 3 expression, caveolae in EM but not caveolae mediated CTxB endocytosis. The manuscript is interesting by itself because it investigates the question in which neuronal cell types caveolae can be found. This question was already studied and discussed previously, however the use of the NT2/D1 cell line is a novel approach and reveals the strong difference in caveolae appearance between astrocytes and neurons.
Major comments:
- The authors determine the expression level of caveolins (Cav1, 2 and 3) by RT-PCR and Western Blot. However, all other caveolae proteins are missing. In particular, Cavin1-3 expression and Immunostaining/ Western Blot should be added as these proteins are essential for the typical caveolae coat and formation of caveolae at the plasma membrane. Also EHD2, EHBP1 and Pacsin2 are important caveolae proteins that could be added to the RT-PCR analysis. This would improve and complete the expression analysis very much and are essential.
- The authors very nicely show the detailed expression of Cav2 and its new variant (fig. 2), however the corresponding protein level experiments are missing in fig. 3. Only Cav1 and 3 are detected by immunostaining and western blot. Why not Cav2? In particular, it would be very interesting to further investigate this new Cav2 variant on the protein level. Many Cav2 antibodies are commercially available and this would improve the manuscript.
- In general, the figures should be improved. In many images the scale bar is missing, also the description of the used cell types or the different fluorescence channels are hidden in the fig. legends which makes it difficult to follow the single sub panels. When showing co-staining of 2 proteins (e.g. fig. 1, 3, 6, 7) the single channels and the merged channels should be placed next to each other , this helps the reader to actually see and evaluate the co-staining. Also enlarged zoomed in images should be added to all co-staining experiments for better evaluation. In fig. 1D for example the staining is basically not visible, and also the costaining with the astrocyte marker is missing, which again makes it difficult to evaluate the experiment. In fig. 3B the costaining with the neuron marker is missing, fig. 6C the colocalization of cav1 and CTxB is not seen in detail, a enlarged image of the plasma membrane would help here. These should be done for all images, and would help to improve the manuscript tremendously.
- The manuscript contains very often the reference to data which is not shown. See line 260, 281, 357, 365, 372, 363, 365, 427, 428, 453, 515. These are very often important controls or additional data and should be included in the manuscript (or supplement) or not discussed in the text.
- Based on the WB, RT-PCR, EM and CTxB data it is very clear that neurons do not contain caveolae, because cav1 is missing and CTxB endocytosis was not observed. However the authors claim in the EM images to detect caveolae (see table 2 and fig. 4). This is the critical point here, and should be investigated better in detail (because it s highly discussed in the caveolae field). However, Cav1 immunogold labeling (or cav2 , cav3) was only done in astrocytes (Fig. 5). Here a detailed analysis of the neurons should be added. The vesicles in fig 4C look very large, and I doubt that these are caveolae vesicles. Again the scale bar is missing which makes it difficult to evaluate.
Minor comments:
- 3H: labeling of the western blot should be added.
- Line 353: “and particularly enriched in the cell processes” what does this means? Which processes , rephrase
- 3G and line 373: show complete western blot membrane, “Western blot analysis of protein lysates revealed a single immunoreactive band of 374 caveolin-1 at ~22 kDa”
- Line 385: include the data which shows cav1 localization to the plasma membrane. Here a co-staining with cav1 and a membrane dye and / or TIRF imaging can be added easily. Also a costaining with cavin1 would improve the results.
- Line 402 “caveolae are expressed” caveolae are not expressed, only the corresponding proteins. (Rephrase)
- Table 2: the number of caveolae are measured, however the unit is not clear (x caveolae / um membrane or um^2 membrane ?) Here, the highest numbers for caveolae are in neurons which is very surprising as cav1 expression is absent. Please add the cav1 / cav2 /cav3 immunogold labeling to be sure caveolae are analysed in the EM images and not other PM invaginations.
- 5B: enlarged images should be added as it is very difficult to see any caveolae
- 7: why are the panels so different for astrocytes and neurons? Again this makes it hard to evaluate. Fig. 7B: Images do not show CTxB uptake into the cells. Rather, CTxB is accumulating at the plasma membrane. Again Cav1 (or Cav2 and Cav3 ) is missing, therefore it cannot be investigated if CTxB and caveolins colocalize.
- Line 474-478: very long and complicated sentence.
- 8: Why are the experiments illustrated in A are performed by flow cytometry but not for experiments in B and C? Why in B and C a plate reader was used? The text indicates that only flow cytometry was used (line 512). Also this analysis did not only measures the endocytosed CTxB but also the accumulated CTxB at the plasma membrane. Again the controls for the NT2/D1 cells are missing. In addition, it should be noted that cholesterol depletion is a very harsh method and it changes the plasma membrane in many different ways which makes it difficult to evaluate if the observed reduced CTxB endocytosis is only caused by caveolae removal. To be exact a cav1 and or cavin1 knockdown experiment (not for this manuscript , but in general) is much more precisely and reveals if caveolae are involved in this process. As the authors also discussed, CTxB uptake is not an optimal assay to evaluate caveolae endocytosis precisely because many other endocytic pathways are involved in the uptake of CTxB.
- Line 532-535: rephrase because the manuscript only investigates caveolae but not the plasma membrane in general
- Line 536: rephrase, “resulting in the development of a functional endocytosis system”.. caveolae are only one out of many endocytic pathways. Also endocytosis also occurs in Nt2/D1 cells, so this sentence is a bit misleading.
- Maybe shorten the discussion a bit, it is very long and in some parts difficult to follow
- Abstract line 25/26: “We have established that NT2/N and NT2/A expressed functional caveolae and caveolins were differentially regulated. “ I think this should be rephrased as it is not supported by the data presented here that neurons contain caveolae and that they are functional. Or add the EM immunogold labeling for caveolins in neurons to clarify this. The CTxB assay in neurons also did not show any caveolae dependence.
- I think this paper should be added to the introduction as it shows the importance of caveolae function in the brain: https://www.nature.com/articles/s41586-020-2026-1, Caveolae in CNS arterioles mediate neurovascular coupling, Chow et al, 2020
Reviewer 3 Report
The manuscript presents a novel overview of the role of caveolin in neuronal and astrocytic-like cell culture models. Overall it is a well-written manuscript, with a very detailed discussion and interpretation of the results. I have a few minor comments (see below), which would make the manuscript more robust and comprehensive for publication.
- The introduction is well-written. It lacks general literature supporting the role of caveolins in glia or astrocytes. The author states that previously published studies have identified the role of caveolin in neurons, but not much in glia or astrocytes. This is important to understand the rationale of the study, as in why the authors decided on pursuing a complex model like neuron-glia mixed culture over glial cultures only. The authors need to include the following manuscript, as it is critically related to the work they are pursuing: Mol Cell Neurosci. 2013 Sep; 0: 283–297.
- In the established cell-culture model, the authors need to evaluate the percentage population of cells that differentiates into neurons from the D1 population.
- In Fig 2, the authors should quantify the expression in different cell types using a quantitative real-time PCR.
- Please provide the full blot with internal controls in figure 3H
- Is there any literature on how does the caveolin expression/patterns change with stress factors like oxidative stress/ nitrosative stress? It will be beneficial to understand the full spectrum of the function of this protein under physiologic as well as stressed conditions. This can also give an indirect overview of caveolin expression and its role during reactive gliosis.
- Since the authors discuss different differentiation conditions under which caveolin expression can be stimulated in neurons in cell line-based models, it will be critical to demonstrate the expression of caveolin through western blotting and qPCR in primary neurons from any species.
Author Response
Reviewer#3
- The introduction is well-written. It lacks general literature supporting the role of caveolins in glia or astrocytes. The author states that previously published studies have identified the role of caveolin in neurons, but not much in glia or astrocytes. This is important to understand the rationale of the study, as in why the authors decided on pursuing a complex model like neuron-glia mixed culture over glial cultures only. The authors need to include the following manuscript, as it is critically related to the work they are pursuing: Mol Cell Neurosci. 2013 Sep; 0: 283–297.
The work by Niesman et al., Mol Cell Neurosci 2013 (reference#15) was cited in line 69. We have added ‘Niesman et. al., have demonstrated that while caveolin-1 expression was increased in activated microglia, it was decreased and redistributed from the plasma membrane to cytoplasmic vesicles in resting microglia [15]’.
- In the established cell-culture model, the authors need to evaluate the percentage population of cells that differentiates into neurons from the D1 population.
It has been previously established by Pleasure et al (1992) that only 4.4 ±0.8% of the initial population become terminally differentiated into NT2/N neurons. We have included this statement in line 257.
- In Fig 2, the authors should quantify the expression in different cell types using a quantitative real-time PCR
Although qPCR could have been performed, we sought to RT-PCR as it is a well-established technique to study the number of specific RNA molecules in a sample. The use of qPCR vs RT-PCR does not change the interpretation of the data. We will certainly add qPCR data for our next manuscript.
- Please provide the full blot with internal controls in figure 3H
Regretfully, the full WB was not saved. Equal amounts of protein (5 µg) was loaded in all gels.
- Is there any literature on how does the caveolin expression/patterns change with stress factors like oxidative stress/ nitrosative stress? It will be beneficial to understand the full spectrum of the function of this protein under physiologic as well as stressed conditions. This can also give an indirect overview of caveolin expression and its role during reactive gliosis.
This topic is discussed in the discussion section starting at line 637, Ref# 67, 68 and 69.
- Since the authors discuss different differentiation conditions under which caveolin expression can be stimulated in neurons in cell line-based models, it will be critical to demonstrate the expression of caveolin through western blotting and qPCR in primary neurons from any species.
Although this data would be nice to generate, it is not the focus of our study. Here we have focused on the use of human neural cells, especially neuron-astrocyte cocultures to recapitulate communication between the two cell types in modulating caveolin proteins.
We would like to thank the reviewer for a thorough review of our manuscript. We believe the corrections made would significantly enhance our publication.
Round 2
Reviewer 2 Report
Line 552
“we have shown for the first 553 time that retinoic acid treatment of NT2/D1 progenitor cells was associated with a remarkable alteration of caveolae-like structures in the differentiated NT2/N neuron-like cells and NT2/A astrocyte-like cells, as evidenced by electron microscopy and changes in the expression of the caveolin proteins (Figures 3 and 4)” and
“Neurons clearly contain caveolae-like structures, we have clarified the statement “We have established that while NT2/A expressed functional caveolae, NT2/N expressed caveolae-like structures and caveolin proteins were differentially regulated in these cells”
– No, the authors haven’t shown that for the NT2/N cells. For the astrocytes they nicely showed the immunogold labeling but in the neurons they only included one EM image without any verification. Again, I have to repeat my previous comments this is a highly critical point here. In general, caveolae are not found in neurons, so if the authors claim that they observe them, the immunogold labeling experiments have to be included, ideally with Caveolin1 and Cavin1 antibody. The observed plasma membrane invaginations in the neurons , and independently, caveolin2 and caveolin3 expression, but this doesn’t mean that the invaginations are caveolae-like structures. Without any immunogold labeling this cannot be claimed.
Comment 1: “We fully agree with the reviewer that it would be nice to generate the data for the expression of cavin proteins, but it is impossible to generate this data set for this manuscript due to the lengthy culture conditions needed to obtain differentiated cells.”
I still think the expression levels of other caveolae proteins should be included, or the focus of the manuscript should be changed to Caveolin expression and not caveolae investigation.
Comment 2: “We have mainly focused on caveolin-1 and 3 in this paper as these are the main proteins involved in caveolae formation and signal transduction. We will certainly pursue the new cav2 variant in a separate study.”
I don’t understand this comment, because in your manuscript you discuss caveolin2 expression including the new caveolin2 variant. So why not including the western blot to further show the different caveolin2 variants on the protein level?
Comment 3:
“We agree with the reviewer that although all the information is present in the figure legends it might be difficult to extract it. We have added the cell types used and estimated the scale bar from magnification. Wherever possible, we have shown the individual channels and merge.”
What does this mean “estimated the scale bar from magnification”? Please can you explain this. The pixel size should be saved in the image information, so by using ImageJ/Fiji you can easily measure the exact image size and scale bar. Please include single channels for Fig. 3C, the reader cannot evaluate the single fluorescence intensities.
“. In Fig 3C, GFAP expression and caveolin-1 is shown, the pattern of caveolin-1 is clear in 3B and 3C and regretfully we could not split the channels as this image was saved as a Tif file”
I don’t understand why the individual channels cannot be shown. When imaging at the microscope the single channels are acquired and saved individually, so this shouldn’t be a problem.
Comment 5:
“We have replaced fig 4C with a new image of neuronal caveolae taken at the same magnification as A, B and D. EM photomicrographs in panels A, B, D were taken at 20,000x magnification, while C was taken at 30,000x magnification and that’s why the neuronal caveolar structures appeared larger. We have added scale bars to all panels in fig 4.”
That’s not the response which was asked (see also first comment line 552). The authors have to prove that in the neurons these vesicles are caveolae by immunogold labeling (accordingly to the astrocytes).
“The antibodies used in our study are very clean and detected only a single band on the western blots. Regretfully, the full WB was not saved.” – That’s not acceptable, all original blots must be saved long-term.
Author Response
Line 552&553: No, the authors haven’t shown that for the NT2/N cells. For the astrocytes they nicely showed the immunogold labeling but in the neurons they only included one EM image without any verification. Again, I have to repeat my previous comments this is a highly critical point here. In general, caveolae are not found in neurons, so if the authors claim that they observe them, the immunogold labeling experiments have to be included, ideally with Caveolin1 and Cavin1 antibody. The observed plasma membrane invaginations in the neurons and independently, caveolin2 and caveolin3 expression, but this doesn’t mean that the invaginations are caveolae-like structures. Without any immunogold labeling this cannot be claimed.
Response: We agree with the reviewer that caveolae are not found in mouse neurons, but to the best of our knowledge, caveolae and caveolin proteins have not been thoroughly investigated in human cells, especially in human neuron-astrocyte cocultures.
As per the reviewer’s suggestion, we have changed the “caveolae-like” structures in neurons to plasma membrane invaginations.
Comment 1: “We fully agree with the reviewer that it would be nice to generate the data for the expression of cavin proteins, but it is impossible to generate this data set for this manuscript due to the lengthy culture conditions needed to obtain differentiated cells.”
I still think the expression levels of other caveolae proteins should be included, or the focus of the manuscript should be changed to Caveolin expression and not caveolae investigation.
Response: We have established the molecular identity of the caveolae in NT2/A, while it remains unknown in NT2/N. Accordingly, appropriate changes have been made in the manuscript.
Comment 2: “We have mainly focused on caveolin-1 and 3 in this paper as these are the main proteins involved in caveolae formation and signal transduction. We will certainly pursue the new cav2 variant in a separate study.”
I don’t understand this comment, because in your manuscript you discuss caveolin2 expression including the new caveolin2 variant. So why not including the western blot to further show the different caveolin2 variants on the protein level?
Response: We would have certainly added the requested western blot data, however we are unable to generate new data due to the lengthy culture conditions needed to obtain NT2N and NT2A.
Comment 3: What does this mean “estimated the scale bar from magnification”? Please can you explain this. The pixel size should be saved in the image information, so by using ImageJ/Fiji you can easily measure the exact image size and scale bar. Please include single channels for Fig. 3C, the reader cannot evaluate the single fluorescence intensities.
Response: Since the images were saved as zvi files, we have used the Carl Zeiss software to add the scale bars, which also uses the pixels from the magnification of the images. It is essentially the same as described by the reviewer.
“In Fig 3C, GFAP expression and caveolin-1 is shown, the pattern of caveolin-1 is clear in 3B and 3C and regretfully we could not split the channels as this image was saved as a Tif file”
I don’t understand why the individual channels cannot be shown. When imaging at the microscope the single channels are acquired and saved individually, so this shouldn’t be a problem.
Response: We have split the green and red channels in Fig 3C.
Comment 5: “We have replaced fig 4C with a new image of neuronal caveolae taken at the same magnification as A, B and D. EM photomicrographs in panels A, B, D were taken at 20,000x magnification, while C was taken at 30,000x magnification and that’s why the neuronal caveolar structures appeared larger. We have added scale bars to all panels in fig 4.”
That’s not the response which was asked (see also first comment line 552). The authors have to prove that in the neurons these vesicles are caveolae by immunogold labeling (accordingly to the astrocytes).
Response: We fully agree with the reviewer.
Caveolin-1 immunogold labeling of NT2N neurons showed no signal, as supported by RNA and protein data. We had hypothesized that in human neurons, caveolin-3 might be sufficient for caveolae formation and therefore had carried out caveolin-3 immunogold labeling. However, the signal was very weak and the results are not publishable.
“The antibodies used in our study are very clean and detected only a single band on the western blots. Regretfully, the full WB was not saved.” – That’s not acceptable, all original blots must be saved long-term.
Response: Point well taken.
Reviewer 3 Report
"Although qPCR could have been performed, we sought to RT-PCR as it is a well-established technique to study the number of specific RNA molecules in a sample. The use of qPCR vs RT-PCR does not change the interpretation of the data. We will certainly add qPCR data for our next manuscript." - A quantifiable PCR read out is needed to understand the relative distribution of the mRNA abundance. Although, it does not change the outcomes, it does impact better understanding and interpretation of the data.
"Regretfully, the full WB was not saved. Equal amounts of protein (5 µg) was loaded in all gels." - This is not acceptable as a rebuttal.
Author Response
Response: Point well taken and we fully agree with the reviewer.
It is impossible to generate new data set for this manuscript due to the lengthy culture conditions needed to obtain differentiated cells. The protocols used for derivation of NT2N and NT2A involves treatment with retinoic acid for 4 weeks followed by a maturation period of 3 weeks. Therefore, it would take us 3-4 months to generate this data.